# Recognizing Students and Detecting Student Engagement with Real-Time Image Processing

**Mustafa Uğur Uçar** [1] and **Ersin Özdemir** [2,*]

1   Havelsan A.Ş. Konya Ofisi 3, Ana Jet Üs K.lığı, Konya 42160, Turkey; mucar@havelsan.com.tr
2   Department of Electrical and Electronics Engineering, Iskenderun Technical University, Iskenderun 31200, Turkey
*   Correspondence: ersin.ozdemir@iste.edu.tr

**Abstract:** With COVID-19, formal education was interrupted in all countries and the importance of distance learning has increased. It is possible to teach any lesson with various communication tools but it is difficult to know how far this lesson reaches to the students. In this study, it is aimed to monitor the students in a classroom or in front of the computer with a camera in real time, recognizing their faces, their head poses, and scoring their distraction to detect student engagement based on their head poses and Eye Aspect Ratios. Distraction was determined by associating the students' attention with looking at the teacher or the camera in the right direction. The success of the face recognition and head pose estimation was tested by using the UPNA Head Pose Database and, as a result of the conducted tests, the most successful result in face recognition was obtained with the Local Binary Patterns method with a 98.95% recognition rate. In the classification of student engagement as Engaged and Not Engaged, support vector machine gave results with 72.4% accuracy. The developed system will be used to recognize and monitor students in the classroom or in front of the computer, and to determine the course flow autonomously.

**Keywords:** computer vision; machine learning; engagement detection; head pose estimation; eye aspect ratio

## 1. Introduction

With the COVID-19 pandemic, there were disruptions in many sectors including education. Young people in Turkey had to continue their education with the courses given online and through television. In face-to-face lessons, teachers can easily monitor if students follow the subject and the course flow is directed accordingly. It is important to determine instantly how much the education has reached to the students and to direct the course accordingly. In this project, it was aimed to recognize the student in e-learning and to measure if the student concentrates on the lesson, to determine and analyze the non-verbal behaviors of the students during the lesson through image processing techniques.

Just like the teacher observes the classroom, the e-learning system must also see and observe the student. In the field of computer vision, subjects, such as object recognition, face detection, face tracking, face recognition, emotion recognition, human action recognition/tracking, classification of gender/age, driver drowsiness detection, gaze detection, and head pose estimation are among the subjects that attract many researchers and are being studied extensively.

Detecting and analyzing human behavior with image processing techniques has also gained popularity in recent times and significant studies have been carried out in different fields.

It is important for students to pay attention to the teacher and the lesson in order to learn the relevant lesson. One of the most important indicators showing that the student pays attention to the lesson and the teacher is that the student is looking towards the teacher.

In this study, it was aimed to find the real-time distraction rates and real-time recognition of students in the classroom or in front of the computer by using images taken from the camera, image processing techniques, and machine learning algorithms. An algorithm has been developed to organize the learning materials in a way to increase interest by providing instant feedback from the obtained data to the teacher or autonomous lecture system. Before going into the details about the study, it will be useful to mention what is student engagement and the symptoms of distraction which are a good indicator to detect student engagement.

The Australian Council of Educational Research defines student engagement as "students' involvement with activities and conditions likely to generate high quality learning" [1]. According to Kuh: "Student engagement represents both the time and energy students invest in educationally purposeful activities and the effort institutions devote to using effective educational practices" [2].

Student engagement is a significant contributor to student success. Several research projects made with students reveal the strong relation between student engagement and academic achievement [3,4]. Similarly, Trowler, who published a comprehensive literature review on student engagement, stated that a remarkable amount of literature has established robust correlations between student involvement and positive outcomes of student success and development [5].

As cited by Fredrickj et al. [6], student engagement has three dimensions: behavioral, cognitive, and emotional engagement. Behavioral engagement refers to student's participation and involvement in learning activities. Cognitive engagement corresponds to student's psychological investment in the learning process, such as being thoughtful and focusing on achieving goals. On the other hand, emotional engagement can be understood as the student's relationship with his/her teacher and friends; feelings about learning process, such as being happy, sad, bored, angry, interested, and disappointed.

Observable symptoms of distraction in students can be listed as looking away from the teacher or the object to be focused on, dealing with other things, closing eyes, dozing off, not perceiving what is being told. While these symptoms can be easily perceived by humans, detecting them via computer is a subject that is still being studied and keeps up to date. Head pose, gaze, and facial expressions are the main contributors to behavioral and emotional engagement. Automated engagement detection studies are mostly based on these features. In order to detect the gaze direction for determining distraction, methods such as image processing techniques, following head pose and eye movements with the help of a device attached to the person's head and processing the EEG signals of the person, were utilized in several studies.

In this study, we focus on behavioral engagement, since its symptoms, such as drowsiness, looking at an irrelevant place can be detected explicitly, and we prefer using image processing techniques for recognizing students and detecting student engagement due to their easy applicability and low cost.

By processing the real-time images of the students in the classroom or in front of the computer captured by a CMOS camera, with the help of image processing and machine learning technologies, we obtain the following results:

1. Face recognition models of the students were created, students' faces were recognized using these models and course attendance was reported by the application we developed;
2. Face detection and tracking were performed in real time for each student;
3. Facial landmarks for each student were extracted and used at the stage of estimating head pose and calculating Eye Aspect Ratio;
4. Engagement result was determined both for the classroom and each student by utilizing head pose estimation and Eye Aspect Ratio.

There are research works and various applications on the determination of distraction in numerous fields, such as education, health, and driving safety. This study was carried out to recognize a student in a classroom or in front of a computer and to estimate his/her attention to the lesson.

The literature review was made under two main titles including the studies conducted on face recognition to identify student participating in the lesson and determining the distraction and gaze direction of the students taking the course.

Studies on Face Recognition:

Passwords, PINs, smart cards, and techniques based on biometrics are used in the authentication of individuals in physical and virtual environment. Passwords, PINs can be stolen, guessed or forgotten but biometric characteristics of a person cannot be forgotten, lost, stolen, and imitated [7].

With the developments in computer technology, techniques that do not require interaction based on biometrics have come to the fore in the identification of individuals. While characteristics, such as face, ear, fingerprint, hand veins, palm, hand geometry, iris, retina, etc., are frequently used in the recognition based on physiological properties, characteristics, such as walking type, signature, and the way of pushing a button, are used in the recognition based on behavioral properties.

There are many identification techniques based on biometrics that make it possible to identify the individuals or verify them. While fingerprint-based verification has reliable accuracy rates, face recognition provides lower accuracy rates due to reasons, such as using flash, image quality, beard, mustache, glasses, etc. However, while a fingerprint reader device is needed for fingerprint recognition and also the person needs to scan his/her fingers on this device, only a camera will be fine to capture images of people. That is why face recognition has become widespread recently.

The first step in computer-assisted face recognition is to detect the face in the video image (face detection). After the face is detected, the face obtained from the video image is cropped and normalized by considering the factors such as lighting and picture size (preprocessing). The features on the face are extracted (feature extraction) and training is performed using these features. By classifying with the most suitable image among the features in the database, face recognition is achieved.

Bledsoe, Chan, and Bisson carried out various studies on face recognition with a computer in 1964 and 1965 [8–10].

A small portion of Bledsoe's work has only been published because an intelligence agency sponsoring the project allowed only limited sharing of the results of the study. In the study, images matching or resembling with an image questioned in a large database formed a subset. It was stated that the success of the method can be measured in terms of the ratio of the number of photos in the answer list to the total number of pictures in the database [11].

Bledsoe reported that parameters in face recognition problem, such as head pose, lighting intensity and angle, facial expressions, and age, affect the result [9]. In the study, the coordinates associated with the feature set in the photos (center of the pupil, inner corner of the eye, outer corner of the eye, mid-point of the hair-forehead line, etc.) were derived by an operator and these coordinates were used by the computer. For this reason, he named his project "man-machine". A total of 20 different distances (such as lip width, width of eyes, pupillary distance, etc.) were calculated from these coordinates. It was stated that people doing this job can process 40 photos per hour. While forming the database, distances related to a photograph and the name of the person in that photograph were associated and saved to the computer. In the recognition phase, distance associated with the photograph in question were compared with the distances in each photograph in the database and the most compatible records were found.

In recent years, significant improvement has been made also in face recognition and some groundbreaking methods have emerged in this field. Now, let us check some of these methods and studies.

The Eigenfaces method uses the basic components of the face space and the projection of the face vectors onto the basic components. It was used by Sirovich and Kirby for the first time to express the face effectively [12]. This method is based of Principal Component Analysis (PCA), which is a statistical approach or the Karhunen–Loève extension. The purpose of this method is to find the basic components affecting the variety in pictures the most. In this way, images in the face space can be expressed in a subspace with lower dimension. It is based on a holistic approach. Since this method is based on dimension reduction, recognition and learning processes are fast.

In their study, Turk and Pentland made this method more stable for face recognition. In this way, the use of this method has also accelerated [13].

The Fisherfaces method is a method based on a holistic approach that was found as an alternative to the Eigenfaces method [14]. The Fisherfaces method is based on reducing the dimension of the feature space in the face area using the Principal Component Analysis (PCA) method and then applying the LDA method formulated by Fisher—also known as the Fisher's Linear Discriminant (FLD) method—to obtain the facial features [15].

Another method widely used in pattern recognition applications is the Local Binary Patterns method. This method was first proposed by Ojala et al. [16,17]. This method provides extraction of features statistically based on the local neighborhood; it is not a holistic method. It is preferred because it is not affected by light changes. The success of this method in face recognition has been demonstrated in the study conducted by [18].

Another method that has started to be used widely in recent years in face recognition is deep learning. With the development of Backpropagation applied in the late 1980s [19,20], the deep learning method emerged. Deep learning suggests methods that better model the work of the human brain compared to artificial neural networks. Most of the algorithms used in deep learning are based on the studies made by Geoffrey Hinton and by the researcher from the University of Toronto. Since these algorithms presented in the 1980s required intensive matrix operations and high processing power to process big data, they did not have a widespread application area in those years.

Convolutional Neural Networks (CNN), which are a specialized architecture of deep learning gives successful results especially in the image processing field and are used more commonly in recent times [21].

It can be said that the popularity of CNN constituting the basis of deep learning in the science world started when the team including Alex Krizhevsky, Ilya Sutskever, and Geoffrey E. Hinton won the "ImageNet Large Scale Visual Recognition Challenge-ILSVRC 2012" organized in 2012 by using this method [22,23].

In recent years, there are successful studies using YOLO $V_3$ (You only look once) algorithm [24], which is an artificial neural network-based, for face detection, and Microsoft Azure (face database), which uses a face API for face recognition [24].

Today, many open-source deep learning libraries and framework, including Tensor-Flow, Keras, Microsoft Cognitive Toolkit, Caffe, PyTorch, etc., are offered to users.

*Studies on Determining Distraction and Gaze Direction*

Correct determination of a person's gaze direction is an important factor in daily interaction [25,26]. Gaze direction comes first among the parameters used in determining visual attention. The role of gaze direction in determining attention level was emphasized in a study conducted by [25] in the 1970s and these findings were confirmed in later studies by [27].

It is stated that the gaze direction defined as the direction the eyes point in space, is composed of head orientation and eye orientation. That is, in order to determine the true gaze direction, both directions must be known [28]. In this study, it was revealed that head pose contributes to gaze direction by 68.9% and to the focus of attention by 88.7%. These results show that head pose is a good indicator for detecting distraction.

In other words, it can be said that the visual focus of attention (VFoA) of a person is closely related to his/her gaze direction and thus the head pose. Similarly, in the study

conducted by [29], head pose was used in determining the visual focus of attention of the participants in the meetings.

When the head and eye orientations are not in the same direction, two kinds of deviations in the perceived gaze direction were stated [30–33]. These two deviations in the literature are named as repulsive (overshoot) and attractive (attractive).

The study conducted by Wollaston, is one of the pioneers in this field [34]. Starting from a portrait in which the eyes is pointing to the front and the head is directed to the left from our perspective, Wollaston draw a portrait with eyes copied from the previous image and the head directed to the right from our perspective and investigated the perceived gaze directions in these two portraits. Although the eyes in the two portraits are exactly the same, the person in the second portrait is perceived as looking to the right. Due to this study, the perception of gaze direction towards the side of the head pose is also called the Wollaston effect as well as the attractive effect [32].

Three important articles were published on this topic in the 1960s [35–37]. The results of two of these studies were investigated and interpreted in different ways by many authors. In the study conducted by Moors et al., the possible reasons for the confusion that causes these two studies to be interpreted in different ways, were discussed [33]. In order to eliminate these uncertainties and clarify the issue, they repeated the experiments performed in both studies and found that there was a repulsive effect in both studies.

Although there are different interpretations and studies, it can be easily said that head pose has a significant effect on the gaze direction.

In their study, Gee and Cipolla also preferred image processing techniques [38]. In the method they used, five points on the face (right eye, left eye, right and left end points of the mouth, and tip of the nose) were firstly determined. By using eye and mouth coordinates, the starting point of the nose above the mouth is found, and the gaze direction was detected with an acceptable accuracy by drawing a straight line (facial normal) from this point to the tip of the nose. This method which does not require much processing power was applied in real time in video images and the obtained results were shared.

Sapienza and Camilleri [39] developed the method of Gee and Cipolla [38] a little further and used 4 points by taking the midpoint of the mouth rather than two points used for the mouth. By using the Viola–Jones' algorithm, face and facial features were detected, the facial features were tracked with the normalized sum of squared difference method and the head pose was estimated from the features followed using a geometric approach. This head-pose estimation system they designed found the head pose with a deviation of about 6 degrees in 2 ms.

In a study conducted in 2017 by Zaletelj and Kosir, students in the classroom were monitored with Kinect cameras, their body postures, gaze directions, and facial expressions were determined, features were extracted through computer vision techniques, and 7 separate classifiers were trained with 3 separate datasets using machine learning algorithms [40]. Five observers were asked to estimate the attention levels of the students in the classroom for each second in their images and these data are included in the datasets and used as a reference during the training phase. The trained model was tested with a dataset of 18 people and an achievement of 75.3% was obtained in the automatic detection of distraction. Again, in another study conducted in the same year by Monkaresi et al. [41], a success of 75.8% was obtained when reports were used during activity and 73.3% success was obtained when the reports after the activities were used in determining the attention of students using the Kinect sensor.

In Raca's doctoral thesis titled "Camera-based estimation of student's attention in class", the students in the classroom were monitored with multiple cameras and the movements (yaw) of the students' head direction in the horizontal plane were determined. The position of the teacher on the horizontal plane was determined by following his/her movements along the blackboard. Then, a questionnaire was applied for each student with 10-min intervals during the lesson, to report their attention levels. According to the answers

of the questionnaire, it has been determined that engaged students are those whose heads are towards the teacher's location on the horizontal axis [42].

Krithika and Lakshmi Priya developed a system in which students can be monitored with a camera in order to improve the e-learning experience [43]. This system processes the image taken from the camera and uses the Viola–Jones and Local Binary Patterns algorithms to detect the face and eyes. They tried to determine the concentration level according to the detection of the face and eyes of the students. Three different concentration levels (high, medium, and low) were determined. The image frames are marked as low concentration in which the face could not be detected; medium concentration in which the face was detected but one of the eyes could not be detected (in cases where the head was not exactly looking forward); high concentration in which the face and both eyes were detected. The developed system was tested by showing a 10-minute video course to 5 students and the test results were shared. According to the concentration levels in test results, parts of course which needed to be improved were determined.

Ayvaz, ve Gürüler also conducted a study and developed an application using OpenCV and dlib image processing libraries aiming to increase the quality of in-class education [44]. Similar to determining the attention level, it was aimed to determine the students' emotional states in real time by watching them with a camera. This system performed the face recognition with the Viola–Jones algorithm, then the emotional state was determined according to the facial features. The training dataset used in determining the emotional state was obtained from the face data of 12 undergraduate students (5 female and 7 male students). The facial features of each student for 7 different emotional states (happiness, sadness, anger, surprise, fear, disgust, and normal) were recorded in this training dataset. A Support Vector Machine is trained with this dataset and emotional states of the students during the lesson were determined with an accuracy of 97.15%. Thanks to this system, a teacher can communicate directly with the students and motivate them when necessary.

In addition to the above studies, there are numerous studies conducted on automatically detecting student engagement and participation in class by utilizing image processing techniques based on the gaze direction determined the student engagement and participation based on their facial expressions [45–51]. In China, a system determining the students' attentions based on their facial expressions was developed and this system was tested as a pilot application in some schools [52].

In the study conducted by [53], the visual focus of attention of the person was tried to be determined by finding his/her head pose using image processing techniques. For this purpose, a system with two cameras was designed in which one camera was facing to the user and the other was facing where the user looks. They developed an application by using OpenCV and Stasm libraries. They tried to estimate head pose of the person in vertical axis (right, left, front) from the images taken from the camera looking directly to the person and which object the person paid attention by determining the objects in the environment from the images taken from the other camera. In order to estimate head pose, the point-based geometric analysis method was used and Euclidean distances between the eye and the tip of the nose were utilized. In the determination of objects, the Sobel edge detection algorithm and morphological image processing technique were used. It was stated that satisfactory results were obtained in determining the gaze direction, detecting objects and which objects were looked at, and the test results were shared.

One of the parameters used in determining the visual focus of attention of a person is the gaze direction. Gaze detection can be made with image processing techniques or eye tracking devices attached to the head. The use of image processing techniques is more advantageous than other methods in terms of cost. However, getting successful results from this method depends on having images of the eye area with sufficient resolution. On the other hand, eye tracking devices are preferred because of their high sensitivity, although they are expensive.

Cöster and Ohlsson conducted a study aiming to measure human attention using OpenCV library and Viola–Jones face recognition algorithm [54]. In this study, images in which faces were detected and not detected were classified by the application developed in this context. It is assumed that if a person is looking towards an object, that person's attention is focused on that object and images, in which no face can be detected because of the head pose, are assumed to be inattentive. The face recognition algorithm was tested with a dataset in which each student is placed in a separate image and looking towards different directions during the lesson. It is ensured with the dataset that the students in the images are centered and there are no other objects in the background. In order to make a more comprehensive evaluation, three different pre-trained face detection models that come with OpenCV Library are used for face detection. The images in the dataset were classified as attentive or inattentive according to the face detection results. It was expressed that this experiment, which is conducted in a controlled environment, yielded successful results.

Additionally, an important part of the studies on detecting distraction is within the scope of driving safety. Driver drowsiness detection is one of the popular research topics in image processing. Tracking the eye and head movements/direction is one of the mostly used methods in the area of driver drowsiness detection. Although there are many studies on this subject in the literature, Alexander Zelinsky is one of the leading names in this field. He released the first commercial version of the driver support system called faceLAB in 2001 [55]. This application performs eye and head tracking with the help of two cameras monitoring the driver and determining where the driver is looking, the driver's eye closeness, and blinking. The driver's attention status is calculated by analyzing all these inputs and the driver is warned before his/her attention level reaches critical levels [56,57].

## 2. Materials and Methods

In this section, hardware and software requirements of our application and the methods used in this study are explained.

### 2.1. Hardware

During developing and running our application for this study, an Intel(R) Core (TM) i7-4770K CPU @ 3.50 GHz computer with 16 GB RAM, Gigabyte NVIDIA GeForce GTX 980 4 GB video card and a Logitech HD Webcam C525 is used.

### 2.2. Software Development Environment and Software Libraries

Windows 10 Pro 64-bit Operating System, Qt Creator 4.2.1 software development environment, and OpenCV 3.3.1, dlib 19.8 software libraries are used.

### 2.3. Methods Used

In the application developed in this study, we used the following methods:

- Histogram of Oriented Gradients (HOG) method was used for face detection;
- Local Binary Patterns (LBP) method was used for face recognition;
- Head pose estimation was performed by solving the Perspective-n-Point (PnP) problem;
- For the classification of "Engaged" and "Not Engaged" students, a Support Vector Machines (SVM) machine learning algorithm was used;
- Functions in dlib library were used for face detection and extracting the facial features and algorithms in OpenCV library were used for face tracking, face recognition, and head pose estimation.

The details of the methods used will be explained in this section according to their orders in the algorithm.

### 2.3.1. Histogram of Oriented Gradients (HOG)

In this method, histograms prepared according to the density changes in the image are used as features [58]. In the method they proposed, an image defined as a $64 \times 128 \times 3$ matrix is represented with a feature vector with 3780 elements [59].

In order to extract HOG features of an image, the image is scaled into the desired dimensions first and it was then filtered with the masks shown in Figure 1 and the gradient images on the x and y axis are obtained. As an alternative to these masks, the Sobel operator can also be used for filtering.

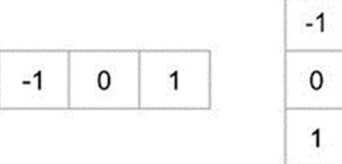

**Figure 1.** Masks used to obtain the gradient images on the *x*- and *y*-axis.

As $g_x$ and $g_y$ refer to the images filtered in *x*- and *y*-axis, the gradient size (*g*) and its direction ($\theta$) can be calculated as in Equation (1):

$$g = \sqrt{g_x{}^2 + g_y{}^2} \quad \theta = \arctan\frac{g_y}{g_x} \tag{1}$$

The density changes of an image in *x*- and *y*-axis (edges and corner) become apparent with the gradient of the image. Gradient images of a sample image on the *x*- and *y*-axis are given in Figure 2. It can be observed that the gradient on the *x*-axis makes the vertical lines clear, while the gradient on the *y*-axis makes the horizontal lines clear.

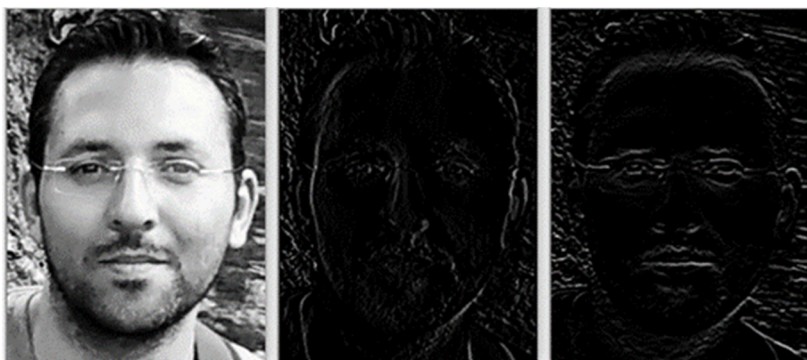

**Figure 2.** Original image (**left**), gradient image on *x*-axis (**center**), gradient image on *y*-axis (**right**).

After obtaining the gradient magnitudes (*g*) and directions ($\theta$) for each pixel, the image is divided into blocks in the second step and the histograms of the gradients for each block are calculated. There are 9 elements in the histogram, including $0°$, $20°$, $40°$, $60°$, $80°$, $100°$, $120°$, $140°$, and $160°$.

### 2.3.2. Local Binary Patterns (LBP)

The Local Binary Patterns method was first proposed by Ojala et al. [16,17]. This method is based on the local neighborhood of pixels which is commonly used in pattern recognition applications. Unlike Eigenfaces and Fisherfaces, which are other methods commonly used in face recognition, the LBP method is not a holistic method, it allows statistical extraction of the features. One of the strengths of LBP is that it is not affected too much by light changes.

In the original version of LBP, the pixel value in the center of blocks in an image with $3 \times 3$ pixel-size is set as the threshold value, and this value is compared with adjacent

pixels individually. In the case when the adjacent pixel value is lower than the threshold value, the new value of the adjacent pixel is assigned as "0", otherwise it is assigned as "1". After this process is completed for 8 adjacent pixels, adjacent pixels are taken sequentially according to the determined direction (clockwise or counterclockwise) and an 8-digit binary number is obtained. The decimal equivalent of this binary number becomes the LBP code of the center pixel. An LBP image is obtained by processing all pixels in the image. An example thresholding process is shown in Figure 3.

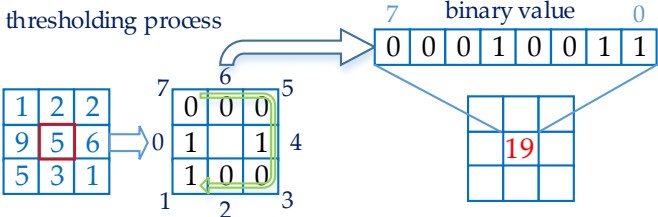

**Figure 3.** Application of the Local Binary Patterns method.

As $(x_m, y_m)$ is the coordinate of the pixel in the center of the 2-dimensional image, $d_m$ is the value of the pixel in the center, and $d_p$ is the value of the adjacent pixel processed, the operation explained above can be expressed as in Equation (2):

$$LBP(x_m, y_m) = \sum_{p=0}^{p-1} 2^p \times f(d_p - d_m)$$

$$f(x) = \begin{cases} 0, & x < 0 \\ 1, & x \geq 0 \end{cases}$$

(2)

It was observed that the original *LBP* method processed with fixed $3 \times 3$ blocks was successful in classifying textures but this method did not give successful results in different scales. In their study, Ahonen et al. developed this method by using it in face recognition applications and demonstrated the success of the method [18]. The new method, which is accepted as the improved version of LBP and also known as Circular LBP, made the current method more stable with two new parameters, radius (r) and pixel count (n), to overcome the scaling problem. Neighborhood examples in the circular LBP are shown in Figure 4.

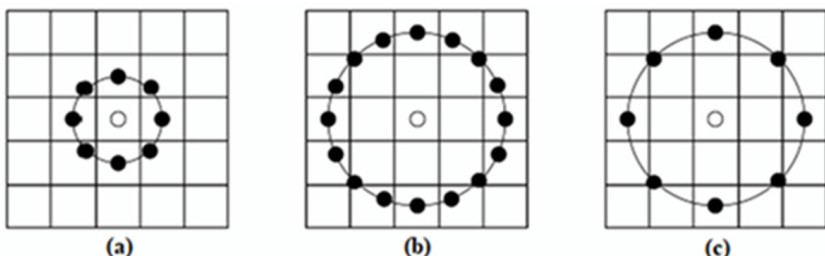

**Figure 4.** Circular LBP examples. (**a**) r = 1, *n* = 8 (**b**) r = 2, *n* = 16 (**c**) r = 2, *n* = 8.

Considering that the LBP radius is r, the LBP adjacent pixels count is $n$, the adjacent pixel to be processed is $p$, the coordinate of the center pixel is $(x_m, y_m)$, the coordinate of the adjacent pixel to be processed is $(x_p, y_p)$ and $p \in n$, and the coordinate of the adjacent pixel for Circular *LBP* can be found as in Equation (3):

$$x_p = x_m + r \cos \frac{2\pi p}{n} \quad y_p = y_m + r \sin \frac{2\pi p}{n}$$

(3)

In case when the coordinate found in this way does not correspond to a certain pixel on the 2-dimensional image, the coordinate can be found by applying a bilinear interpolation as in Equation (4):

$$f(x,y) = [1-x \quad x] \begin{bmatrix} f(0,0) & f(0,1) \\ f(1,0) & f(1,1) \end{bmatrix} \begin{bmatrix} 1-y \\ y \end{bmatrix} \tag{4}$$

After finding decimal LBP codes for pixels in each block and forming the new LBP image, LBP histograms for each block in the new image are formed. By adding the value of each pixel in the block being processed, the histogram for that specific block is obtained. This process is repeated for all blocks in the image. A sample LBP image and the combined histogram of this image are shown in Figure 5.

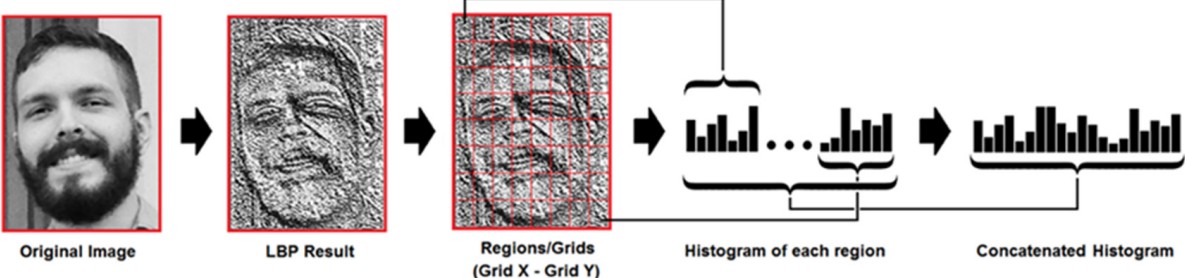

Original Image　　　　LBP Result　　　Regions/Grids　　Histogram of each region　　Concatenated Histogram
　　　　　　　　　　　　　　　　　(Grid X - Grid Y)

**Figure 5.** Forming the LBP histogram, the image was taken with the permission of Kelvin Salton do Prado from his article titled "Face Recognition: Understanding LBPH Algorithm" [60].

### 2.3.3. Support Vector Machines (SVM)

The Support Vector Machines machine-learning method is a supervised learning method that can perform classification and regression on a tagged training dataset.

The elements that separate the classes and are closest to the decision boundary are called the support vectors.

After training, a new point to be given to the system is classified by the SVM model according to its position with respect to the decision boundary.

Our n-element training set that can be separated linearly, can be expressed as $\left(\vec{x}_1, y_1\right), \ldots, \left(\vec{x}_n, y_n\right)$. Here, $\vec{x}_i$ indicates each element in our training set, $y_i$ indicates which class this element belongs to which is either $-1$ or 1. $\vec{w}$ refers to the weight vector (normal vector perpendicular to the decision boundary), b refers to bias and the decision boundary for a linear SVM can be expressed as in Equation (5):

$$\vec{w} \times \vec{x} - b = 0 \tag{5}$$

Two parallel lines (negative and positive hyperplanes) that are used to find the decision boundary and separate the classes from each other are as in Equation (6):

$$\vec{w} \times \vec{x} - b = \quad 1$$
$$\vec{w} \times \vec{x} - b = -1 \tag{6}$$

These explanations can be seen more clearly in Figure 6.

SVM has a linear kernel; so, it has been developed that it can make classification in cases where linear classification is not possible. In this case, the data are moved to a higher-dimensional space and classified using the "Kernel Trick" method. For this purpose, various kernel functions can be used with SVM. Commonly used ones of these are Linear, Polynomial, Gaussian Radial Basis Function, and Sigmoid kernel functions.

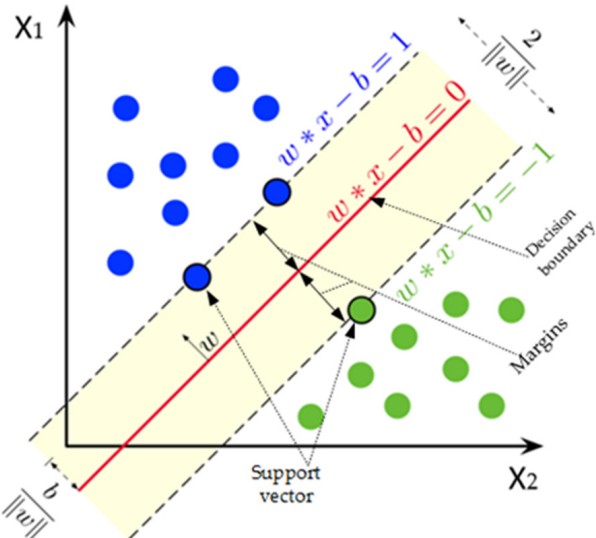

**Figure 6.** Decision boundary and support vectors in SVM.

### 2.3.4. Solving PnP problem and Finding Euler Angles

Perspective-n-Point or simply the PnP problem is a problem of predicting the pose of a calibrated camera if the coordinates of n points of an object are known in both 2-dimensional and 3-dimensional space. With the solution of this problem, rotation and translation values of the objects in front of the camera can be estimated with respect to the camera. We can express the rotational movement of a 3-dimensional (3D) object relative the camera with the rotation angles around the $x$-, $y$-, and $z$-axis. The same is valid for the translation. In order to known the translation of a 3D object with respect to the camera, it is necessary to know the translation amount according to the $x$-, $y$-, and $z$-axis. In this case, the movement of an object according to the camera, which is the combination of rotation and translation values, is called pose in the literature and requires 6 values to be known.

In the image processing field, many methods have been proposed to solve this problem, which attracts the attention of researchers. From these solutions, the solution of Gao et al. [61], known as P3P, the solution of Lepetit et al. [62], known as EPnP, and the PnP solutions proposed by Schweighofer and Pinz [63] stand out as non-iterative solutions. The solution proposed by Lu et al. [64] is one of the iterative solutions with high accuracy.

The solvePnP function which comes with OpenCV library provides many solutions of this problem to researchers. In order to use solvePnP() function, one requires

- 3-dimensional coordinates of the points of the object;
- 2-dimensional coordinates for the same points in the image of the object;
- Parameters specific to the camera used.

After solvePnP function is run, it produces a rotation vector and a translation vector as output parameters.

The Euler angles are named after Leonhard Euler. These angles introduced in 1776 by Leonhard Euler are the three angles describing the orientation of a rigid object with respect to a fixed coordinate system [65]. Euler angles also known as pitch, yaw, and roll angles for a head example are illustrated in Figure 7.

In order to calculate the Euler angle, the Rodrigues rotation formula [66] is applied to the rotation vector obtained from the solvePnP function and the rotation matrix is obtained. For this transformation, the Rodrigues function that comes ready with the OpenCV library, can be used. Using the obtained rotation matrix, the Euler angles corresponding to the rotation matrix are calculated by applying a transformation as described in [67].

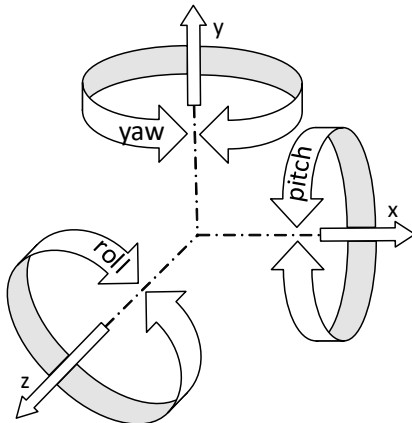

**Figure 7.** Euler angles.

2.3.5. Detecting Eye Aperture

Eye aperture is calculated by the ratio known as Eye Aspect Ratio in the literature. Soukupová and Cech successfully used the Eye Aspect Ratio to detect blink moments [68].

Eye aperture is an important parameter in order to determine whether the student is sleepy or distracted. When the eyes are narrowed or closed, the eye aspect ratio approaches to 0. This situation is seen in Figure 8.

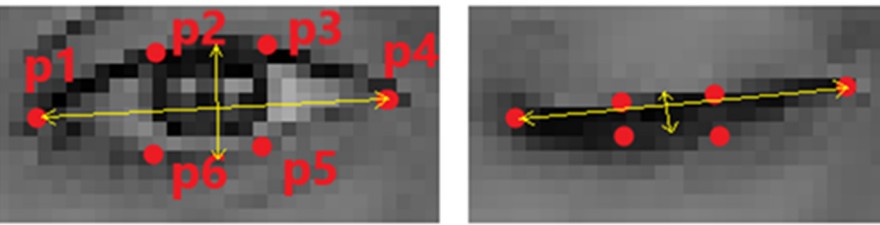

**Figure 8.** The states of the landmarks according to the open and closed position of eyes.

By utilizing the Euclidean distance formula and the coordinates of the landmarks seen in Figure 8, the Eye Aspect Ratio can be calculated as in Equation (7):

$$EAR = \frac{\|p2 - p6\| + \|p3 - p5\|}{2 \times \|p1 - p4\|} \tag{7}$$

*2.4. Dataset Used*

In order to test the success of face recognition, head pose estimation, and student engagement detection in the developed application, the UPNA Head Pose Database was used [69].

UPNA Head Pose Database

This database is composed of videos of 10 different people, including 6 men and 4 women. It was designed to be used in studies about head tracking and head pose. In the database, there are 12 video files for each person. People in these videos moved (translation) and rotated (rotation) their heads along the *x*-, *y*-, and *z*-axis. The videos in the database were recorded at 1280 × 720 resolution at 30 fps in MPEG-4 format and each video is composed of 10 s, that is, 300 frames [70].

Along with the database, the text documents containing the measurement values obtained through the sensors for each video are also shared. Three types of text documents are shared for each video:

2-dimensional measurement values (*\*_groundtruth2D.txt*): It consists of 300 rows, one row for each frame. Each row has 108 columns separated by the TAB character. In these 108 columns, there are x and y coordinate values ($x_1$ $y_1$ $x_2$ $y_2$ $\cdots$ $x_{54}$ $y_{54}$) of 54 facial landmarks.

3-dimensional measurement values (*\*_groundtruth3D.txt*): It consists of 300 rows, one row for each frame. Each row has 6 columns separated by the TAB character. In these columns, the translation amounts of the head along the *x-*, *y-*, and *z*-axis are given in mm and the rotation angles are given in degrees. It has been reported that these values are measured with high precision with the help of a sensor attached to the user's head [70]. These measurement values are as follows:

a. Translation amount of the head along the *x*-axis ($T_x$);
b. Translation amount of the head along the *y*-axis ($T_y$);
c. Translation amount of the head along the *z*-axis ($T_z$);
d. Roll, which is the rotation angle along the *z*-axis (roll);
e. Yaw, which is the rotation angle along the *y*-axis (yaw);
f. Pitch, which is the rotation angle along *x*-axis (pitch).

3-Dimensional face model (*\*_model3D.txt*): This file contains x, y, and z coordinates of 54 landmarks on the face for the corresponding person [70].

The sample images of the videos in the database are seen in Figure 9.

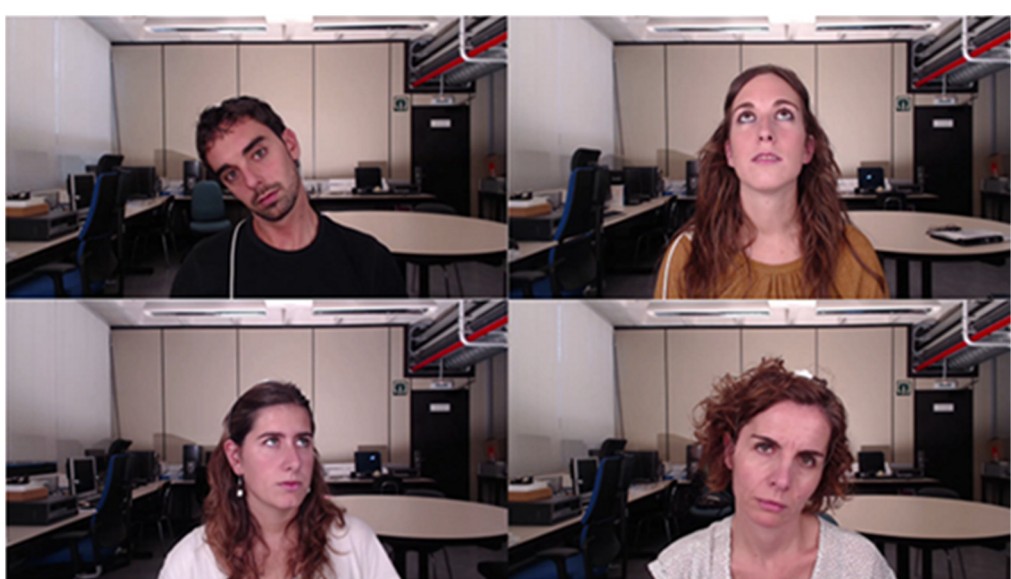

**Figure 9.** Sample images taken from the UPNA Head Pose Database.

## 3. Developed Algorithm of Student Engagement Classification Software (SECS)

Student Engagement Classification Software is designed to determine distraction in real time according to head poses of students and also recognize them:
SECS software consists of two parts. These sections are:

1. Face Recognition Model Preparation Module;
2. Main module.

### 3.1. Face Recognition Model Preparation Module

The algorithm of SECS software is developed to prepare the face recognition model as below.

As it can be seen in the flowchart (Figure 10), before starting the recording process, a student name must be entered. Then, 50 photos are taken for each student at 5-s intervals through the webcam and the student is asked to pose as different as possible during the recording process (such as looking at different directions, rotate his/her head, posing with and without glasses, with eyes closed and opened, different facial expressions, etc.). After

the recording process is completed for all students, the face recognition model is created using the captured photos.

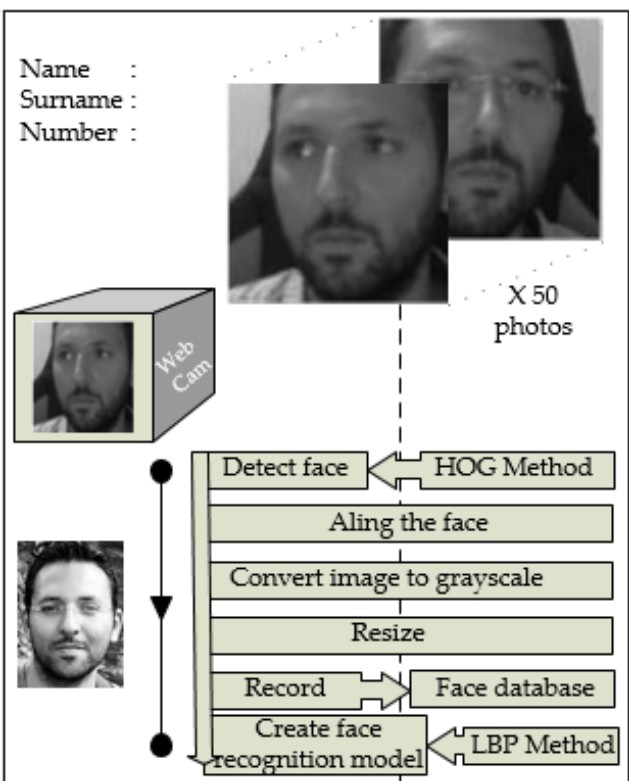

**Figure 10.** Flowchart of face recognition model preparation module of SECS.

*3.2. Main Module*

In the main part of SECS, face detection from the real-time images taken from the camera, facial landmarks extraction, face recognition, head pose estimation, and student engagement classification processes are carried out.

In addition to these, if the student is not far from the camera and the eye area can be properly determined with image processing, eye aperture is also calculated to increase the success of student engagement classification. This parameter is added optionally in order for the program to be used in e-learning environments.

The flow diagram of the main module of SECS software is shown in Figure 11. The methods used in the flowchart in the figure are as follows. Histogram of oriented gradients were used for face detection, local binary patterns for face recognition, PnP for head pose estimation, and SVM for classification.

"Engaged" or "Not Engaged" labels that appear on the screenshots are the real-time student engagement classification results generated by the SVM machine learning algorithm used in the application for each frame.

In addition to the real-time tracking of the distraction results, it is also possible to record and report the results in a certain period of time. After the "Start Attention Tracker" button is clicked, the application recognizes the faces detected in each frame and records the student engagement classification results produced by SVM for the relevant people. In addition, the total distraction rates for the detected faces are displayed instantly on the screen. The mentioned total distraction results are shown in Figure 12.

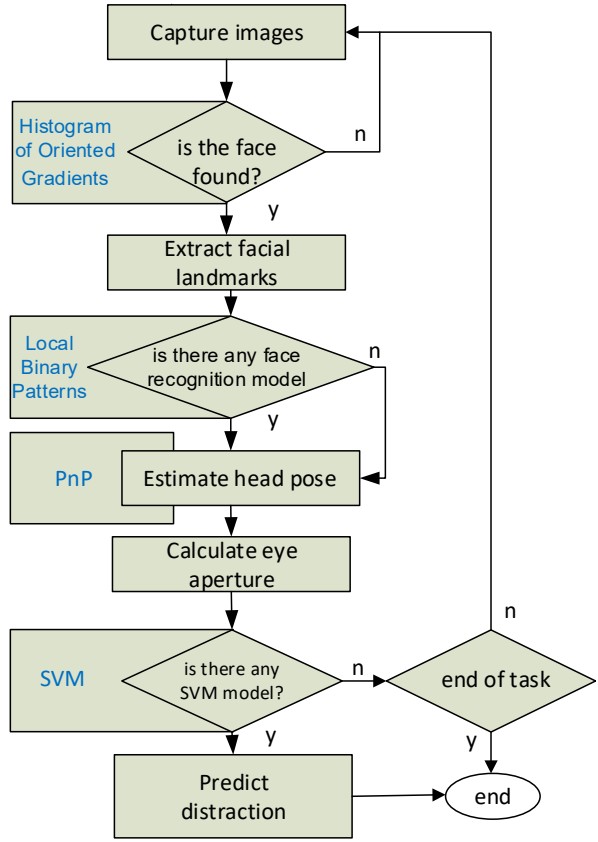

**Figure 11.** Flowchart of main module of SECS.

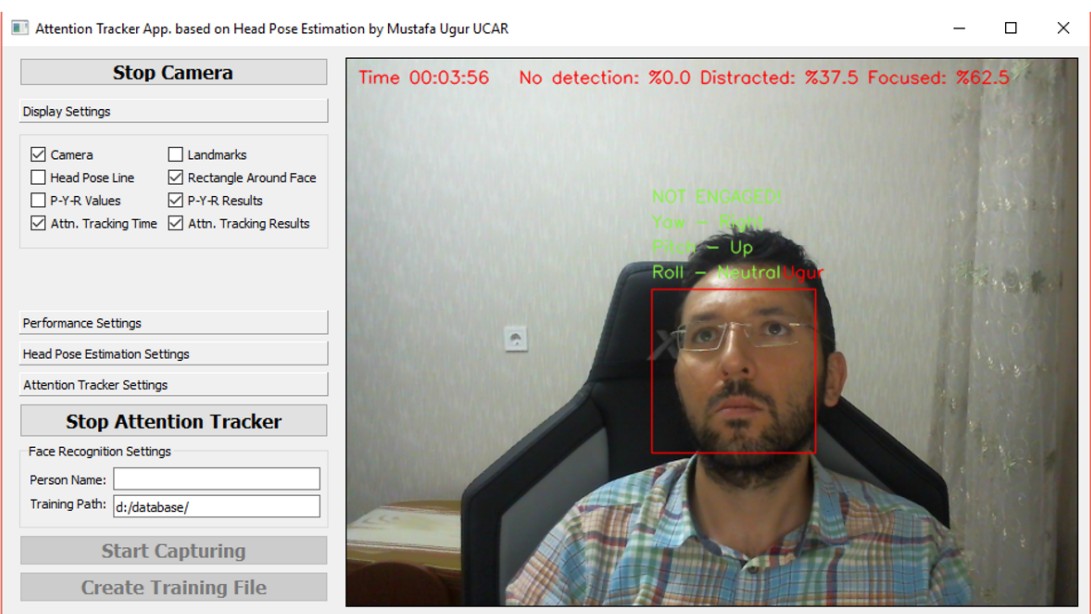

**Figure 12.** SECS main module—instant total distraction results.

In the screenshot in Figure 12, the faces determined by the camera for 3 min and 56 s are seen to be focused (engaged) within the specified time with the rate of 62.5%. Screenshots of the SECS software are given in Figure 13.

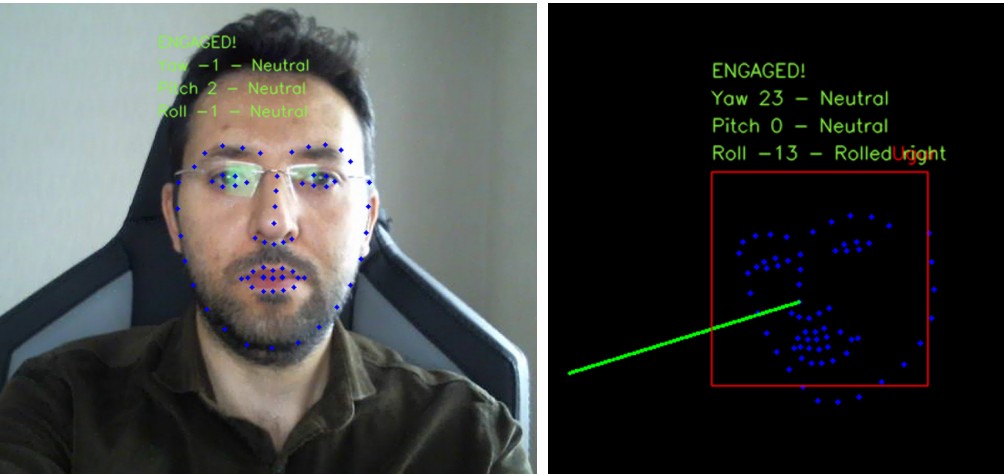

**Figure 13.** SECS main module—various screenshots.

The reporting process is stopped when the "Stop Attention Tracker" button is clicked and the student engagement classification results are recorded in the results.csv file. In this file, the name of the recognized person and the information on how many frames that person is "Engaged" and "Not Engaged" during the follow-up period are included. In the case when the face detection process is performed but the student is not recognized, student engagement classification results of the person are shown in a line named "default" instead of the student's name. A sample .cvs file (result.csv) opened in Excel is shown in Table 1.

**Table 1.** SECS main module—student engagement classification results.

| PersonName | Engaged | NotEngaged |
|:---:|:---:|:---:|
| Ugur | 4301 | 336 |
| Berna | 4214 | 423 |

## 4. Results

The results obtained from this application developed to classify student engagement during a course will be investigated under 3 categories. Results of face recognition, head pose estimation, and evaluation of the whole system will be discussed, respectively.

### 4.1. Results Related to Face Recognition

In order to measure the success of the developed application on face recognition, the UPNA Head Pose Database was used. For this purpose, face alignment, converting to grayscale, and resizing processes were applied to 50 face pictures of 10 students derived from the video images of each student in the database, and these photographs were labelled and recorded with the student's name to be used in the preparation of the face recognition model. By training with LBP, Fisherfaces and eigenfaces face recognition algorithms in the OpenCV library with the labelled photos, the face recognition models were created for each method. Then, using the trained LBP, Fisherfaces, and eigenfaces face recognition models, the face recognition process was performed in each frame of each video in the UPNA Head Pose Database and predicted labels for each frame were recorded.

When the predicted labels are compared with the real label values, the LBP method achieved the recognition with the highest rate of 98.95%. The average face recognition accuracy rates for all people in UPNA Head Pose Database occurred as 98.69% in the Fisherfaces method, 94.57% in the eigenfaces method. The details are shown in Table 2. The LBP method was preferred as the face recognition algorithm giving the best results in the developed application.

**Table 2.** The results of the studied face recognition algorithms as percentages.

|  | LBP % | Fisherface % | Eigenfaces % |
|---|---|---|---|
| User_01 | 98.69 | 99.86 | 90.13 |
| User_02 | 100 | 100 | 99.25 |
| User_03 | 96.36 | 97.58 | 90.22 |
| User_04 | 98.97 | 96.16 | 95.58 |
| User_05 | 100 | 100 | 96.22 |
| User_06 | 99.52 | 99.75 | 96.52 |
| User_07 | 99.08 | 100 | 95.77 |
| User_08 | 99.80 | 99.88 | 97.38 |
| User_09 | 99.83 | 96.36 | 94.11 |
| User_10 | 97.30 | 97.33 | 90.55 |
| Average | 98.95% | 98.69% | 94.57% |

*4.2. Results of Head Pose Estimation*

Head pose estimation in the developed application was obtained with the solution of the Perspective-n-Point problem explained previously. In order to determine the success of the part related to the head pose estimation, head pose angles, calculated by the developed application for each frame (each video is composed of 300 frames) of each video (there are a total of 120 videos in the database), were estimated and recorded. The recorded results are compared with the reference values of head pose angles (pitch, yaw, and roll) that come with the UPNA Head Pose Database and were measured with sensors for each frame of each video. By grouping the comparison results to the persons in the database, the mean absolute deviations of Euler angles were determined and the mean absolute deviations from ground-truth data in pitch, yaw, and roll angles were calculated as 1.34°, 4.97°, and 4.35° respectively. The details are shown in Table 3.

**Table 3.** Comparison of the angles taken from the UPNA Head Pose Database with the angles calculated by our algorithm.

|  | Pitch° | Yaw° | Roll° |
|---|---|---|---|
| User_01 | 1.65° | 2.68° | 1.94° |
| User_02 | 1.10° | 2.39° | 1.52° |
| User_03 | 1.09° | 5.77° | 2.94° |
| User_04 | 1.40° | 3.08° | 1.95° |
| User_05 | 1.24° | 6.50° | 4.36° |
| User_06 | 1.76° | 4.56° | 4.52° |
| User_07 | 1.24° | 4.02° | 2.85° |
| User_08 | 1.33° | 9.50° | 6.24° |
| User_09 | 0.96° | 6.23° | 5.70° |
| User_10 | 1.59° | 4.94° | 11.44° |
| Average | 1.34° | 4.97° | 4.35° |

*4.3. Results Related to the Overall System*

Two separate test scenarios were conducted to measure the success of SECS application.

In the first scenario, it was emphasized if a student engagement classification could be made independent of the machine learning algorithms. For this purpose, the developed application has been modified, the threshold values for pitch, yaw and roll angles were determined. Then, the student was accepted as "Not Engaged" when he/she does not look at the camera (at least one of the pitch, yaw, and roll angle is lower than the threshold). Since the webcam used has insufficient viewing angle, a 4-minute test was conducted with 2 students at the same time. In each minute of these 4 min, the students followed the instructions given to them. For example, when Student1 was asked to look at the camera during the 3rd minute, Student 2 was asked to look in other directions. With the instructions given to Student 1 and Student 2, the student engagement classification process

was initiated and the process was finished at the end of the 4th minute. The instructions given (on the left) and the obtained results are shown in Table 4.

**Table 4.** Overall success of SECS—first scenario.

|  | Student 1 | | Student 2 | |
|---|---|---|---|---|
|  | **Instruction** | **Result** | **Instruction** | **Result** |
| **1. Minute** | 100% | 96% | 100% | 93% |
| **2. Minute** | 50% | 50% | 50% | 45% |
| **3. Minute** | 100% | 88% | 0% | 0% |
| **4. Minute** | 70% | 66% | 35% | 30% |

In the second scenario, a test was carried out with machine learning algorithms. For this purpose, we created a student engagement dataset by using the UPNA Head Pose Database. We extracted random 100 frames for each person from the UPNA Head Pose Database and created a dataset that consists of a total of 1000 images. Five human labelers annotated each image as (0)—"Not Engaged" or (1)—"Engaged". We measured consistency between the labelers based on Fleiss' kappa measure [70], which is a statistical approach for assessing the interrater reliability of more than two raters. Percent agreement and Fleiss' kappa values of our five labelers calculated 0.95 and 0.85, respectively. These values indicate there is strong agreement between labelers. Then, we labelled each image as majority decision of our labelers.

Sample images in our labelled dataset are shown in Figure 14.

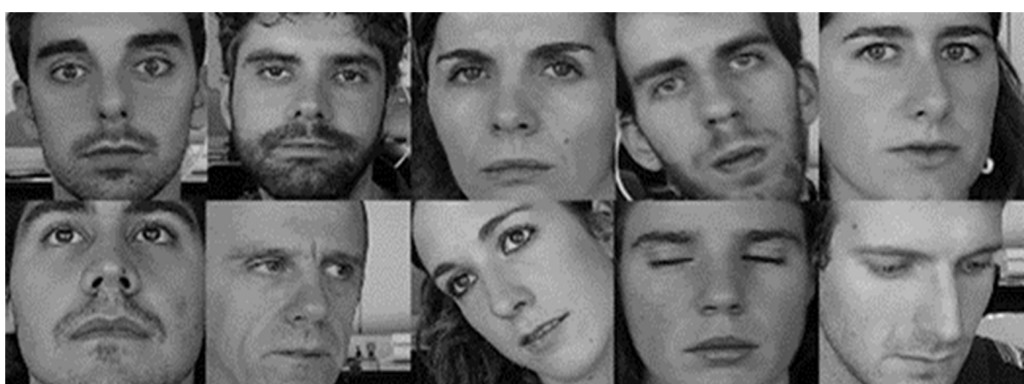

**Figure 14.** Sample photos from the dataset formed to test the overall success of SECS (first line: "Engaged"; second line: "Not Engaged").

After the labelling process is completed, head pose angles and eye aperture ratios in each of these 1000 photos were calculated and recorded as described in Section 3. For each photo saved, an attribute vector in the form of:

$$v = [pitch, yaw, roll, eye\_aspect\_ratio, label]$$

was obtained and half of the 1000 photos was determined as training set and the other half was determined as the test set.

Random Forest, Decision Tree, Support Vector Machines, and K Nearest Neighbor machine learning algorithms in the OpenCV library were trained with the prepared training set, and these models were tested over the test set.

The most successful result in student engagement classification was obtained with Support Vector Machines with an accuracy rate of 72.4% and, thus, the use of SVM was preferred in the student engagement classification part of the developed application. The details regarding the obtained results are shown in Table 5.

**Table 5.** Overall success of SECS—second scenario (by using estimated Euler angles).

|  | Accuracy Rate % |
| --- | --- |
| SVM | 72.4% |
| K Nearest Neighbor | 71.6% |
| Random Forest | 70.6% |
| Decision Tree | 70.0% |

In addition, a second test for the same scenario was carried out on the training and test set prepared by using Euler angles measured with sensors that come with the UPNA Head Pose Database instead of the estimated Euler angles. As a result of this test, the most successful results were obtained with the Random Forest machine learning algorithm with 78.6% accuracy rate. Details regarding the obtained results are shown in Table 6.

**Table 6.** Overall success of SECS—second scenario (by using Euler angles measured with sensors).

|  | Accuracy Rate % |
| --- | --- |
| SVM | 76.4% |
| K Nearest Neighbor | 77.0% |
| Random Forest | 78.6% |
| Decision Tree | 77.8% |

## 5. Conclusions

With the application developed for this study:

- A total of 50 random photo frames were taken for each person from videos of 10 people in the UPNA Head Pose Database; these photos were saved in separate folders with the names of those people and a face recognition model was prepared using these photos. The prepared face recognition model was tested using all videos in the UPNA Head Pose Database and a 98.95% face recognition success was achieved with the LBP method.
- Head pose angles were estimated for 120 videos each of which consists of 300 frames in the UPNA Head Pose Database using image processing techniques. The obtained pitch, yaw, and roll angles were compared with the angle values measured precisely by the sensors. The estimated pitch, yaw, and roll angles were obtained with mean absolute deviation values of $1.34°$, $4.97°$, and $4.35°$, respectively.

In the literature, 75.3% success was reported in the model trained with the data obtained with the Kinect One sensor, which is used to determine the distraction of students, and 75.8% success in another study.

In this study, using 1000 photos taken from the videos from the UPNA Head Pose Database, a distraction database was prepared. Pitch, yaw, roll, and eye aperture rates were calculated for each photo and feature vectors were formed. Each photo in the dataset was labelled as "Engaged" or "Not Engaged" by 5 labellers. Fleiss' kappa coefficient was calculated as 0.85 (high reliability) in order to determine the reliability of the labelling process. While assigning labels to each photo in the dataset, the majority of decisions were taken as the basis. Half of the labelled dataset was used in the training of machine learning algorithms and the other half was used in testing. In the conducted tests, the SVM machine learning algorithm showed 72.4% success rate in determining distraction.

Considering the results of this study conducted on student engagement classification, it is seen that the results obtained using a simple CMOS camera gave similar results with the studies conducted with the Kinect sensor. In addition, the developed system is more useful in terms of recognizing the student and associating the student engagement result with the student.

In this study, algorithms were developed for face detection and recognition, estimation of head pose, and classification of student engagement in the lesson, and successful results were obtained.

Moreover, it was seen that the viewing angle and distance were limited due to the camera used in the developed system. This is especially effective for the success of the face recognition algorithm. In order for the system to provide more successful results in the classroom environment, it is planned to conduct in-class tests using more professional, wide-angle cameras and using additional light.

A labelled dataset consisting of 1000 photos was prepared for the training and tests of machine learning algorithms that will determine student engagement. Expanding this dataset will make the results more successful.

The feature vector used in training student engagement classifiers consists of 5 parameters. In addition to these parameters, adding features, such as eye direction, emotional state of the student, and body direction, is predicted to be effective in increasing the success of student engagement classification.

We continue to work on algorithms that guide the teaching method of the course by determining the course topics and weights, taking into account the characteristics of the students. To give a brief detail, data from this study are one of the input parameters of the fuzzy logic algorithm we use to determine the level of expression during the lesson. Students in front of the computer or in the classroom who are engaged in the lesson are counted. The obtained value and the time spent in that lesson are entered into the fuzzy logic algorithm as input data. From here, it is decided at which level the lecture of the course, which is prepared at different levels, will be chosen. The student's interest is tried to be kept during the lesson. Our work on this issue continues.

**Author Contributions:** Conceptualization, E.Ö.; methodology, E.Ö. and M.U.U.; software, M.U.U.; validation, M.U.U.; formal analysis, E.Ö.; investigation, E.Ö. and M.U.U.; resources, M.U.U.; data curation, M.U.U.; writing—original draft preparation, E.Ö.; writing—review and editing, E.Ö. and M.U.U.; visualization, E.Ö. and M.U.U.; supervision, E.Ö.; project administration, E.Ö.; funding acquisition. All authors have read and agreed to the published version of the manuscript.

**Funding:** This research received no external funding.

**Institutional Review Board Statement:** In this study, no data were collected from the students by camera or any other way. In order to test the success of face recognition, head pose estimation and student engagement detection in the developed application, the UPNA Head Pose Database was used [64].

**Informed Consent Statement:** The person in the photograph in Figures 2, 10, 12 and 13 is my graduate student, Mustafa Uğur Uçar, with whom we work in this study. That is why I attached the informed consent form for subjects. The people in the photos in Figures 9 and 14 are the photos of the people in the UPNA Head Pose Database. There are also photographs of these people on the relevant website. This database is licensed under a Creative Commons Attribution-Non-Commercial-ShareAlike 4.0 International License. For this reason, I could not add the informed consent form for subjects in Figures 9 and 14.

**Data Availability Statement:** Data available on request due to privacy. The data presented in this study are available on request from the corresponding author.

**Conflicts of Interest:** The authors declare no conflict of interest.

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
