# Peer review of "Recognizing Students and Detecting Student Engagement with Real-Time Image Processing"

_electronics, doi:10.3390/electronics11091500_

Round 1
Reviewer 1 Report
The manuscript did not change much or I see a wrong version. Conclusions must be improved, it shall not include literature citations, discussions. Those parts please move to results and discussion. Conclusions are expected to highlight the major findings, suggestions etc.
Deep details of known image processing algorithms are not scientific achievements. I believe the content perfectly fits to a short communication or case study, provided that journal can publish such type of papers.
Author Response
Dear Reviewer,
First of all, thank you very much for your criticism.
The purpose of repeating the references in the Conclusion was to emphasize the results obtained using assistive devices and the results we obtained only with CMOS camera and open-source software. We made the change you suggested and the necessary corrections. References and the explanation in the conclusion section have been edited.
The following paragraph has been added to the conclusion section.
"In the literature, 75.3% success was reported in the model trained with the data obtained with the Kinect One sensor, which is used to determine the distraction of students and 75.8% in another study."
Best regards,
Note: The editing made in the literature and conclusion sections are written in a dark blue tone in the text.

Reviewer 2 Report
The authors have improved the manuscript based on comments from reviewers in the previous submission.
Author Response
Thank you very much for your evaluation.
Note: English and spelling were checked again and some corrections were made.
This manuscript is a resubmission of an earlier submission. The following is a list of the peer review reports and author responses from that submission.
Round 1
Reviewer 1 Report
The topic is interesting, authors used camera to analyse students' attention and focus on the lesson. Methods are developed and introduced to achieve this goal. The manuscript needs editing to meet guidelines by means of structure and content can be more concise in method description and present more details in results.
The novelty of the scientific work is below average. Face/object recognition with proposed technique is already existing such as Yolo, and eye tracking also have even commercial solutions such as Tobii. The face, direction, eye and expression detection/estimation is solved and commercially available with OMRON HVC (Human Vision Component) using single camera for up to 35 persons. The most common eye tracking solutions and their usage in sensory analysis and behaviour analysis are missing from literature review. Basically, the scientific part is already solved, but ignored and even not cited.
The main value of the presented work is the usage of technique, but behaviour analysis is very small part and not defined with sufficient details. For example, when student takes notes of the lesson, it will be recognised "not engaged" falsely because not staring into the camera.
There are structural changes compared to guidelines, I believe authors shall correct:
- featured application L10-17 before abstract is strange, like a pre-abstract to highlight the highlights.
- introduction is followed by related works, what is a second introduction. Please merge them into a single one. There are overlaps as well, due to separation.
English needs minor adjustments, such as:
- "distance education" might be "distance learning" or "remote education" or similar, more commonly used term.
- Eye Aspect Ratios is not common parameter, it is basically eye shape description and blink detection parameter. Bit confusing until it is introduced.
- abstract states accuracy was compared to other methods but no details are provided (L33-34), please remove
- introduction part has very few references, and are obviously missing when authors mention several studies in the topic or specific technique.
- Please edit in-text citation, because "in/of/by [99]" shall be "in/by Author et al. [99]" and all similar (see L80 etc.)
- the camera shall be defined other than "web cam". Please use specific definition instead of everyday slang. The device is a CMOS HD camera.
- please add author such as other publications in L283-290.
- please check manuscript for singular/plural present/past mixtures. Especially work done should be in past tense.
- L657: "ground truth" please use another term to cite reference values.
There are some listings, which needs more information to clarify or remove strange items:
- L176-179: LDA is not a dimension reduction tool, it is used for classification
- L197: NN layers for training and classification do not exist. The input/output can be used for training and classification, validation.
- L220-221: TensorFlow, Cognitive Toolkit are not libraries, but include them.
Specific comments:
- L475-476: SVM has linear kernel, please modify statement
- Figure 13 is a table. Please make a table or a screenshot of your own software.
- reported unit in Table 1,4,5 is %, please indicate
- Table 2 is not reporting success as mentioned in caption, but error. Please modify.
- Table 3 might be better if organized by students and consecutive columns compare instructions and measurements.
- Conclusions is mainly summary of results, including literature as well. Please improve conclusions.
Reviewer 2 Report
The contents of this manuscript are interesting but my main concerns are as follows.
- Paper and related work reviewing should be focused on the topics mention in lines 104-110. Some introductions about CNN in lines 193-221 may be removed or significantly reduced because CNN is not used in the proposed method.
- Too many contents are in the related works lines 111-364 and should be well abbreviated.
- In Tables 1-5, "success" should be replaced by more feasible words, such as "average face recognition accuracy rates" in Table 1 for easier understanding.
- The number of subjects is too few. Are authors possible to collect data by themselves for training or validation?
- The influence of environmental factors such as lighting on the measurement accuracies should be discussed.
Reviewer 3 Report
- There are too much explanations about the importance of problems in introduction. First explain the problems briefly, then mention the relevant research and their shortcomings, and finally say what you have done and what problems you have solved others.
- In the second part of the related work, in the research on face recognition, the author should focus on describing the local binary method he uses, and the rest can be simply omitted without spending a lot of time on it.Part 2.1 can appropriately put forward their own views, and the references need to be updated to add some documents in recent three years.
- Chapter 3 does not need to introduce each method in detail, but simply mention it, focusing on how to judge students "engaged" or "not engaged" according to these methods.
- Line 368: the software and hardware configuration is introduced in the experiment part, and the database in Section 3.4 is introduced in the experiment part.
- LBP and SVM are not chosen because of the good experimental results. This choice is discussed from the perspective of the algorithm itself.
- Figure 2.1 in line 392 should be Figure 3.1. Line 393 y ax is should be y-axis. Figure 3.5 is placed directly below line 452. Figure 3.9 pictures and labels are placed on the same page. In Figure 4.1, it is align, not align. It is recommended to make the picture horizontal, and do not occupy one side of the picture and words. The font on figure 4.3 is blurred. The picture and table are not centered. The color and font of the picture are not uniform.
- There is no experimental comparison of different databases.
- The screenshots of the experimental results show more recognition results under different postures.
- More recent referencesshould be cited.
- More complex scenarios should be considered, such as judging whether to bow your head without listening or taking notes.
- Some format problems need attention, such as Figure 3 The icon question of 9 should be put together with the figure, not on a separate page.
- This paper can be described in detail the flow chart in Figure 4.2 below. The judgment link of is there any face recognition model is unclear.
- The reason for the unrecognized face problem proposed in part 4.2 is not analyzed in the summary, or the next solution is proposed.
- The experimental workload is too small, and the classification results of students' participation (Fig. 4.4) can not fully explain the advantages of the work.
Round 2
Reviewer 1 Report
The manuscript has been improved in the second round. In my opinion it is a methodological study, not a practical application as it would be expected based on the title and abstract. It has useful parts introducing how image processing methods work with details. Unfortunately the results are limited to prove methods reached comparable success to those in literature (face recognition, gaze direction detection) but application in online learning or classroom monitoring is still weaker.
Finally, the contribution to the field is small by means of new scientific information. Maybe title and abstract can change to emphasize methodology approach/development.
English needs minor editing to justify grammar mistakes (singular, plural).